# Electrochemical deposition of N-heterocyclic carbene monolayers on metal surfaces

Einav Amit[1,2,6], Linoy Dery[1,2,6], Shahar Dery[1,2], Suhong Kim[3], Anirban Roy[4], Qichi Hu[4], Vitaly Gutkin[2], Helen Eisenberg[1,5], Tamar Stein[1,5], Daniel Mandler[1,2], F. Dean Toste [3] & Elad Gross [1,2✉]

*N*-heterocyclic carbenes (NHCs) have been widely utilized for the formation of self-assembled monolayers (SAMs) on various surfaces. The main methodologies for preparation of NHCs-based SAMs either requires inert atmosphere and strong base for deprotonation of imidazolium precursors or the use of specifically-synthesized precursors such as NHC(H)[HCO$_3$] salts or NHC–CO$_2$ adducts. Herein, we demonstrate an electrochemical approach for surface-anchoring of NHCs which overcomes the need for dry environment, addition of exogenous strong base or restricting synthetic steps. In the electrochemical deposition, water reduction reaction is used to generate high concentration of hydroxide ions in proximity to a metal electrode. Imidazolium cations were deprotonated by hydroxide ions, leading to carbenes formation that self-assembled on the electrode's surface. SAMs of NO$_2$-functionalized NHCs and dimethyl-benzimidazole were electrochemically deposited on Au films. SAMs of NHCs were also electrochemically deposited on Pt, Pd and Ag films, demonstrating the wide metal scope of this deposition technique.

[1] Institute of Chemistry, The Hebrew University, Jerusalem 91904, Israel. [2] The Center for Nanoscience and Nanotechnology, The Hebrew University, Jerusalem 91904, Israel. [3] Department of Chemistry, University of California, Berkeley, CA 94720, USA. [4] Bruker Nano Surfaces Division, 112 Robin Hill Road, Santa Barbara, CA 93117, USA. [5] The Fritz Haber Center for Molecular Dynamics Research, The Hebrew University, Jerusalem 91904, Israel. [6] These authors contributed equally: Einav Amit, Linoy Dery. ✉email: elad.gross@mail.huji.ac.il

**N**-Heterocyclic carbenes (NHCs) are molecular ligands characterized with strong affinity to metals[1,2]. The strong NHC–metal interaction enabled the formation of stable self-assembled monolayers (SAMs) of NHCs on metals[3–21], metal-oxides[22,23] and semimetals[24]. The wide chemical-tunability of NHCs led to utilization of NHC-based SAMs as biosensors[10,25,26], molecular probes for surface reactivity[27–29], and co-catalysts[3,30–33].

Thus far, NHC-based SAMs have been mostly prepared on metallic surfaces, and specifically on Au surfaces, by two approaches: (i) base-induced deprotonation of imidazolium salt precursors[5,7,17,26,34] and (ii) annealing of NHC(H)[HCO₃] salts[10,13,15,21] or NHC–CO₂ adducts[5,11,14,15] under vacuum conditions (Fig. 1). The deposition process of imidazolium salt precursors with halide ions (Cl⁻, Br⁻, I⁻) is conducted in THF under an anhydrous environment with a strong base, such as potassium tert-butoxide (KO^tBu), for deprotonation and carbene formation[5,7,17,26]. Although this deposition approach has been widely utilized for the preparation of NHC-based SAMs, it has a number of inherent drawbacks: First, an anhydrous environment is required since residual water can quench the active carbene. Moreover, base and solvent residues remain on the surface following liquid-deposition and limit the formation of well-ordered monolayers[16]. Finally, the deposition process requires high concentration of imidazolium salt (~1–10 mM) and extended deposition time (>12 h).

The use of an inorganic base for deprotonation and active carbene formation can be circumvented by using NHC(H) [HCO₃] salts or NHC–CO₂ adducts as masked precursors to the free carbene (Fig. 1)[5,10,11,13–15,21]. Annealing of these precursors under vacuum conditions facilitates the formation and evaporation of an active carbene that can be anchored on metal surfaces. This approach excludes liquid or base residues from the surface, thereby allowing the formation of well-ordered monolayers[10,16].

However, this approach also has several disadvantages: First, NHC–CO₂ adducts and imidazolium carbonate salts precursors require specific preparation, which includes separation steps and ion exchange processes, respectively[10]. Additionally, various functional groups are incompatible with the imidazolium carbonate synthesis, which limits the preparation of chemically addressable NHC-based SAMs. Finally, the deposition technique involves annealing of the precursors and evaporation of the active carbene toward the metal surface. These steps restrict the use of high molecular-mass or temperature-sensitive precursors. Interestingly, NHC-based SAMs have been prepared from imidazolium carbonate salts that have been immersed in alcohols (Fig. 1)[10]. This approach overcomes the need for elevated temperatures, but induces solvent residues on the surface and requires higher concentration of precursors (10 mM).

While the above deposition processes provide useful syntheses of many addressable NHC-based SAMs, the above-mentioned limitations inspired us to consider complementary methods that might avoid some of their restrictions. Herein, we demonstrate an electrochemical approach for the preparation of NHC-based SAMs, in which deprotonation of the imidazolium salt is electrochemically induced (Fig. 1). The electrochemical (EC) deposition utilizes the localized formation of hydroxide ions in proximity to the electrode surface, induced by water reduction under a negative potential (−1 V), for deprotonation of the imidazolium salt (Fig. 2)[35–37]. The proximity between the active carbene and metal electrode enables the formation of NHC-based SAMs under ambient conditions and in the presence of water.

It should be noted that electrochemical reactions that permit organic layers to be attached to conducting solid substrates were previously demonstrated[38]. For example, aromatic organic layers were formed on conductive or semiconductive surfaces following electrochemical reduction of aryl diazonium salts[39,40]. However, utilization of the electrochemical deposition approach toward surface anchoring of NHCs was not previously demonstrated,

Herein, 1,3-bis(2,4-dinitrophenyl)-imidazole (NO₂-NHC)[16,17,27] was used as a model system for addressable carbenes and was EC-deposited on various metal surfaces. Quantitative analysis revealed that EC deposition induced monolayer formation of NO₂-NHCs on Au films with higher surface density and improved chemical stability than those prepared by base-induced deprotonation. The higher surface density and improved chemical stability of EC-deposited SAMs were connected with the fact that during EC deposition a small and constant concentration of active carbenes is formed near the electrode. This proximity provides a short time frame for the sequential deprotonation and surface-anchoring steps, thus limiting the competitive adsorption

**a. Previous works**

X⁻ = Halide anion

R = Mes, t-Bu, i-Pr, Me...
R' = H or –C₄H₄–

**b. This work**

Addressable Imidazolium: R = 2,4-dinitrophenyl, R' = H, X = Br⁻
Benzimidazolium: R = Me, R' = –C₄H₄– , X = I⁻
TBATFB = Tetrabutylammonium tetrafluoroborate

**Fig. 1 Methodologies for preparation of NHCs-based SAMs. a** NHC-based SAMs formation is mainly facilitated by deprotonation of imidazolium salt precursors with inorganic base or by using specifically synthesized precursors such as NHC(H)[HCO₃] salts or NHC–CO₂ adducts. **b** In this work an alternative approach is demonstrated in which SAMs of NHCs are prepared on metal surfaces (Au, Pt, Pd, and Ag) by electrochemically induced deprotonation of 1,3-bis(2,4-dinitrophenyl)-imidazolium bromide and 1,3-dimethyl-benzimidazolium iodide salt precursors.

Addressable Imidazolium: R = 2,4-dinitrophenyl, R' = H
Benzimidazolium: R = Me, R' = –C₄H₄–

**Fig. 2 Suggested mechanism for EC deposition of imidazolium on Au film.** EC deposition mechanism for 1,3-bis(2,4-dinitrophenyl)-imidazolium and dimethyl-benzimidazolium on Au-coated Si electrode.

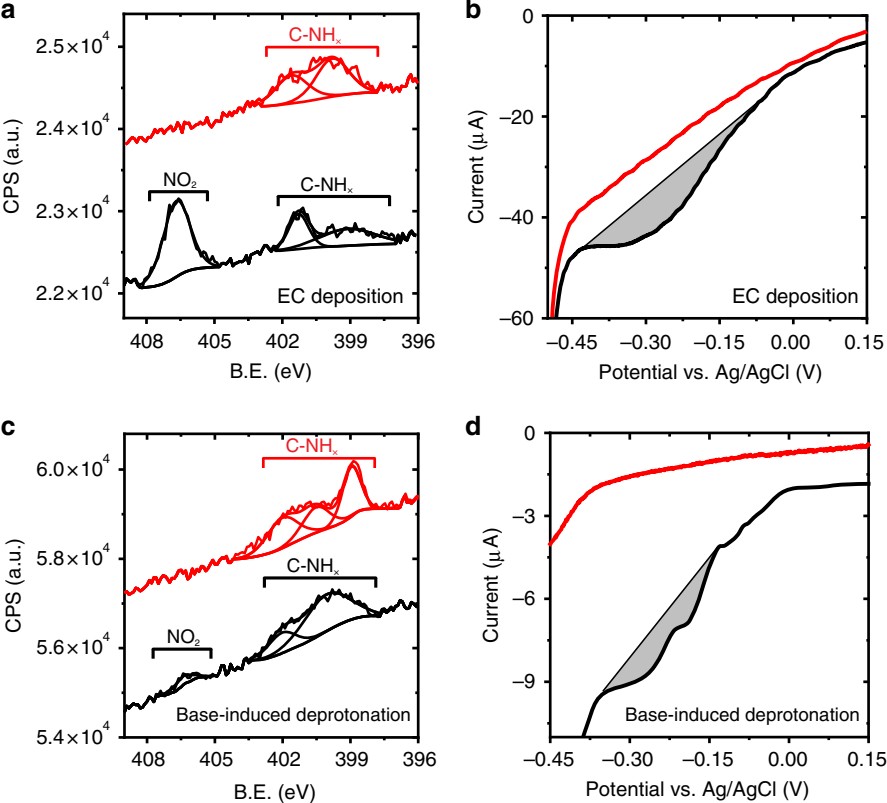

**Fig. 3 Spectroscopic measurements of NO₂-NHCs that were prepared on Au film by EC deposition and base-induced deprotonation.** N1s XPS and LSV measurements of NO₂-NHCs which were deposited on Au films by EC deposition (**a** and **b**, respectively) and base-induced deprotonation (**c** and **d**, respectively). Measurements were performed prior (black-colored) and after (red-colored) one cycle of LSV. LSV conditions: 0.1 M HCl and scan rate of 0.1 V/s.

of carbene and Br⁻ on the Au surface. The wide metal scope of the EC deposition approach was demonstrated and, in addition to Au, SAMs of NO₂-NHCs were prepared on Pd, Pt, and Ag films. The wide NHC scope of the EC deposition approach was demonstrated as well and, in addition to NO₂-NHCs, SAMs of 1,3-dimethyl-benzimidazole were prepared on Au films.

## Results

### Spectroscopic analysis of EC-deposited monolayer of NHCs on Au film

Nitro-functionalized NHCs (NO₂-NHCs) were electrochemically (EC) deposited on Si-supported Au film. In the EC deposition process, hydroxide ions were formed in proximity to the Au-coated Si electrode by applying a negative potential (−1 V) that led to water reduction (Fig. 2). The hydroxide ions function as a base for deprotonation of imidazole cations, enabling active carbenes formation in proximity to the electrode surface. These carbenes were self-assembled on the electrode's surface as identified by N1s XPS measurements (Fig. 3a, black-colored spectrum).

The N1s XPS signal of EC-deposited NO₂-NHCs (Fig. 3a, black-colored spectrum) was constructed of two distinctive peaks, located at 405–408 and 397–403 eV and correlated to NO₂ and C-NH$_x$ species, respectively[41]. The low-energy N1s XPS peak was fit by two Gaussians, centered at 399.4 and 401.3 eV, which were assigned to the amine (N–H) and carbene nitrogen, respectively[17]. The NO₂:CNH$_x$ peaks area ratio was 1.5:1, which is smaller than the stoichiometric 2:1 ratio of NO₂-NHC and indicates that a fraction of the nitro groups were reduced upon their deposition. This conclusion is validated by the presence of an amine-correlated feature in the XPS signal (centered at 399.4 eV).

Electroreduction of the nitro groups in surface-anchored NO₂-NHCs provides a chemical handle for quantitative analysis of the surface density of NHCs, based on the well-documented mechanism of electroreduction of aromatic nitro compounds[42,43]. Linear sweep voltammetry (LSV) of EC-deposited NO₂-NHCs revealed a reduction peak at −0.05 to −0.40 V, correlated to reduction of -NO₂ groups (Fig. 3b, black-colored voltammogram). Similar electroreduction patterns were previously reported for molecules that were functionalized with di- and tri-nitro groups[44,45]. The electroreduction peak was not detected in a consecutive LSV measurement (Fig. 3b, red-colored voltammogram), indicating that the -NO₂ groups were fully reduced during the first electroreduction cycle.

The nitro-to-amine electroreduction was identified as well in the N1s XPS signal (Fig. 3a, red-colored spectrum). The high binding energy peak (405.5 eV), which was correlated to NO₂ species, was not probed after the first LSV cycle. The elimination of this peak was coupled with an increase in the area of the low-binding energy Gaussian in the XPS signal, which was correlated to amine. The noticeable changes in the N1s XPS spectrum and LSV voltammogram following one cycle of LSV indicate that nitro-to-amine electroreduction was facilitated in NO₂-NHCs that were deposited on the Au surface.

### Spectroscopic analysis of NHC monolayer prepared by base-induced deprotonation

The properties of EC-deposited NO₂-NHCs were compared to that of NO₂-NHC SAMs that were prepared using an inorganic base (KO$^t$Bu) for deprotonation of the imidazolium salt under inert conditions (Fig. 1). The N1s XPS signal of Au-anchored NO₂-NHCs, prepared by base-induced deprotonation, showed two distinctive peaks (Fig. 3c, black-

colored spectrum); however, the ratio of the two peaks, correlated to $NO_2$:$CNH_x$ ratio, was 0.07. This value is more than an order of magnitude smaller than the value measured for EC-deposited $NO_2$-NHCs and indicates that most of the nitro groups were reduced during the base-induced deposition process. The deteriorated $NO_2$:$CNH_x$ ratio is consistent with the highly reactive nature of the base-induced deposition approach[16].

LSV measurement of $NO_2$-NHCs that were surface-anchored by base-induced deprotonation showed a much shallower electroreduction peak (Fig. 3d, black-colored voltammogram) in comparison to the peak detected for EC-deposited $NO_2$-NHCs. The presence of a shallower electroreduction peak correlates with the XPS results and shows that most of the nitro groups were reduced during the deposition process.

The high-energy peak in the N1$s$ XPS signal of $NO_2$-NHCs that were prepared by base-induced deprotonation was eliminated after one LSV cycle (Fig. 3c, red-colored voltammogram). Additionally, the low-energy peak in the N1$s$ XPS signal became wider and a dominant feature was identified at 398.9 eV. The detection of a peak at this energy, which is at lower energy than the expected amine peak position, signals that the electroreduction was coupled with partial decomposition of surface-anchored NHCs. A similar decomposition pattern was previously identified for $NO_2$-NHCs that were prepared by base-induced deprotonation and anchored on Pt (111)[16]. The detection of a decomposition peak in a monolayer that was prepared by base-induced deprotonation and exposed to electroreducing conditions shows its deteriorated chemical stability in comparison to that of EC-deposited monolayer.

**Comparative analysis of SAMs prepared by the two deposition techniques.** The surface density of EC-deposited $NO_2$-NHCs was quantified by analysis of the electroreduction peak of the –$NO_2$ groups and was determined to be $(2.3 \pm 0.7) \times 10^{-11}$ mol cm$^{-2}$ (see Supplementary Methods for additional details). Thus, the average surface area for a single surface-anchored $NO_2$-NHC molecule was determined to be $7 \pm 2$ nm$^2$/molecule. Analysis of the influence of EC deposition duration on the surface density of $NO_2$-NHCs showed that a maximum surface density is reached after 5 min of deposition (Supplementary Fig. 1). Extending the electrodeposition duration beyond this point did not noticeably change the surface density of $NO_2$-NHCs.

The surface density of $NO_2$-NHCs that were prepared by base-induced deprotonation was $3.8 \times 10^{-12}$ mol cm$^{-2}$, as quantified by analysis of the electroreduction peak that was detected in LSV measurements. However, this analysis is biased by the fact that most of the nitro groups in $NO_2$-NHCs that were prepared by base-induced deprotonation were already reduced upon their deposition (Fig. 3c, black-colored spectrum). A comparison of the N1$s$/Au4$f$ XPS peaks area ratios revealed 3-fold higher values for EC-deposited NHCs than that of NHCs that were prepared by base-induced deprotonation (Supplementary Table 1). Based upon this ratio it can be calculated that the surface density of $NO_2$-NHCs that were prepared by base-induced deprotonation was $(0.8 \pm 0.2) \times 10^{-11}$ mol cm$^{-2}$.

The higher surface density of EC-deposited $NO_2$-NHCs was connected with lower surface concertation of competitive adsorbates, as identified by XPS measurements. XPS analysis of Au surfaces on which $NO_2$-NHCs were prepared by EC deposition and base-induce deprotonation showed N:Br:K atomic ratios of 1:0.3:0 and 1:1.25:0.4, respectively (Supplementary Table 1 and Supplementary Fig. 2). An inverse correlation was therefore identified between the surface density of NHCs and that of bromide and potassium, indicative of a competitive surface-adsorption process between these species. XPS measurements did

not detect F1$s$ signals on the Au surface on which $NO_2$-NHCs were EC-deposited, demonstrating that electrolyte residues were not adsorbed on the Au surface during EC deposition (Supplementary Fig. 2).

DFT simulations identified that the optimal surface density of $NO_2$-NHCs in a closely packed monolayer was $1.2 \times 10^{-10}$ mol cm$^{-2}$ (Supplementary Fig. 3). Thus, the calculated surface density was 5-fold higher than that of the experimental value. The difference between the experimental and calculated surface density can be linked with the competitive adsorption of bromide and carbene on the Au surface and to the strong interaction of $NO_2$-NHCs with the Au surface that hindered the formation of a dense monolayer in which all surface-anchored molecules are well aligned.

Thus, integration of XPS and LSV results identified that higher surface density and improved chemical stability were achieved by EC deposition of $NO_2$-NHCs. The higher surface density and improved chemical stability of EC-deposited monolayer were attributed to the following factors: (i) Milder deprotonation conditions and (ii) formation of small and constant concentration of carbenes in proximity to the metal surface. These two factors minimized the competitive adsorption of bromide on the surface and the deformation of NHCs upon their surface-anchoring.

**The EC deposition mechanism of NHCs.** Various control experiments were conducted to validate the EC deposition mechanism. Reduction and oxidation cycles of $Fe(CN)_6^{3-}$/$Fe(CN)_6^{4-}$ on the bare and $NO_2$-NHC coated Au electrode showed that no passivation of the electrode was induced following EC deposition (Supplementary Fig. 4), thus excluding multilayer formation by EC deposition. Spectroelectrochemistry measurements demonstrated that imidazolium deprotonation is facilitated only once negative potential ($-1$ V) is applied and $H_2O$ was added to the solution (Supplementary Fig. 5). Similarly, XPS measurements revealed that surface-anchoring of $NO_2$-NHCs was not achieved without water addition or with a lower voltage of $-0.5$ V (Supplementary Fig. 6).

The influence of water concentration on the EC-deposited yield was studied (Supplementary Fig. 7). It was identified that the surface density of NHCs was 4-fold lower once water concentration was decreased from 50 to 5 mM. The surface density of NHCs was not changed once water concertation was increased to 150 mM, demonstrating the self-limited process of monolayer formation by EC deposition. However, higher water concertation induced undesired oxidation reactions within the surface-anchored NHCs. The results of these experiments validate our hypothesis that water reduction led to the formation of a basic environment that facilitated imidazolium deprotonation and the following surface-anchoring of carbene.

The feasibility for EC deposition is based on the fact that hydroxide ions, which are formed by water electroreduction, will function as a base for deprotonation of the imidazolium salt (Fig. 2). The p$K_a$ of 2,4 dinitrophenyl-imidazolium was measured (see Supplementary Methods for additional details and Supplementary Figs. 8–10) and was found to be equal to p$K_a$ = $10.49 \pm 0.02$. The pH in the vicinity of electrode during EC deposition was estimated to be pH = 12.54 (see Supplementary Methods for additional details). Thus, the p$K_a$ of the imidazolium salt is lower than the pH on the electrode, enabling deprotonation of the imidazolium salt by water reduction.

The stability of the EC-deposited $NO_2$-NHCs was studied following exposure to 25 and 50 cycles of cyclic voltammetry ($-0.5$ V to 1 V vs Hg/Hg$_2$SO$_4$). N1$s$ XPS measurements did not reveal noticeable changes in the surface density of $NO_2$-NHCs after 25 cycles (Supplementary Fig. 11). However, the surface

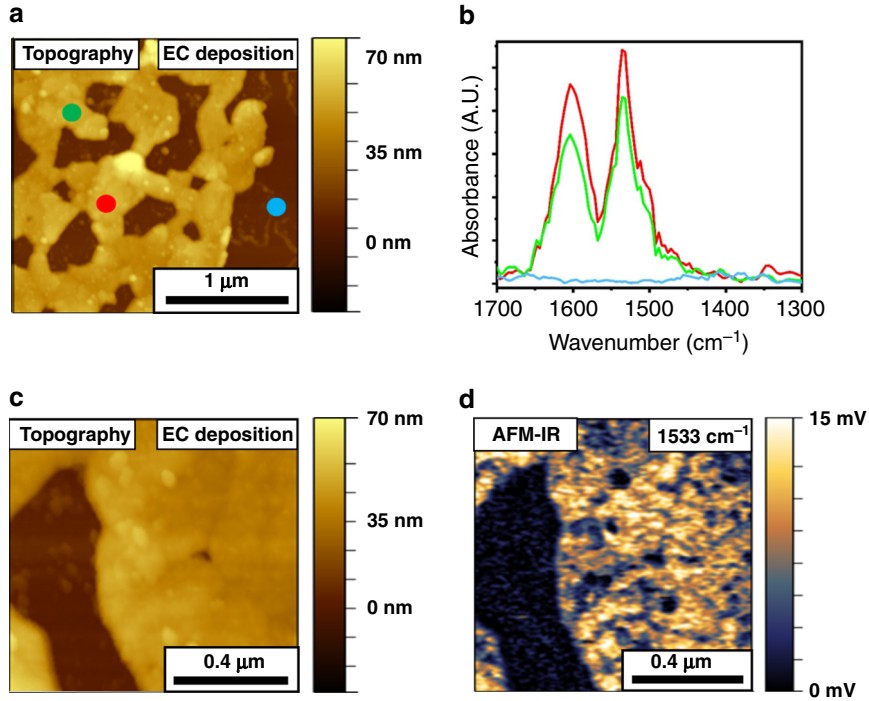

**Fig. 4 AFM-IR measurements of EC-deposited NO₂-NHCs on Au film.** AFM topography (**a**) and AFM-IR point spectra measurements (**b**) following EC deposition of NO₂-NHCs on a patchy Au film that was deposited on Si electrode. Colored circles in **a** mark the local IR measurement positions and the measured IR spectra are shown in **b** with identical color-coding. AFM topography image at higher magnification and the corresponding AFM-IR image at 1533 cm⁻¹ are shown in **c** and **d**, respectively.

density was 5-fold lower after 50 cycles, indicating that electro-induced desorption has occurred.

**High spatial resolution IR mapping of NO₂-NHCs monolayers.** AFM-IR measurements[16,46,47] were performed to complement the ensemble-based measurements and provide high spatial resolution analysis of the distribution and chemical properties of NO₂-NHCs monolayers that were prepared by EC deposition and base-induced deprotonation. AFM-IR measurements provide both structural and chemical information at the nanoscale with a spatial resolution of ~20 nm. These capabilities make it a superb technique for analysis of the averaged distribution and chemical functionality of NHCs on surfaces. The AFM-IR measurements were conducted on a patchy Au film that was evaporated on a Si wafer in order to map the averaged distribution of NHCs on the Au surface and probe leaching of NHCs onto the Si surface.

Figure 4a shows a topographic map of the Si substrate (brown-colored) and the patchy Au film (50–70 nm height, gold-colored) on which NO₂-NHCs were EC-deposited. AFM-IR measurements were conducted on several points across the Au film and the bare Si surface. Colored dots in Fig. 4a mark the locations in which AFM-IR measurements were performed and the measured IR spectra are shown in Fig. 4b with identical color-coding. The spectra measured on the gold surface (red- and green-colored dots in Fig. 4a and red- and green-colored spectra in Fig. 4b) show signals at 1533 and 1603 cm⁻¹, correlated to asymmetric N–O and aromatic C=C vibrations, respectively[28]. Vibrational signals were not detected on the bare Si substrate (blue-colored spectrum in Fig. 4b), demonstrating the selective adsorption of NHCs on the Au surface. Interestingly, stronger vibrational signals were identified on flatter areas, correlated to higher surface density of NO₂-NHCs on these sites (Supplementary Fig. 12).

The AFM-IR spectrum of surface-anchored NO₂-NHCs was compared with the ATR-IR spectrum of the imidazolium salt precursor (Supplementary Fig. 13). Three main peaks were detected in the ATR-IR spectrum of the salt precursor, located at 1340, 1536, and 1608 cm⁻¹ and correlated to the symmetric and asymmetric N–O vibrations and aromatic C=C vibration, respectively. IR spectrum of the imidazolium salt was also deduced by DFT calculations and showed peaks at similar positions to those detected in the ATR-IR spectrum (Supplementary Fig. 13). The infrared reflection absorption spectrum of surface-anchored NO₂-NHCs showed peaks at similar positions to those detected by ATR-IR[17]. The positions of the dominant vibrational peaks of imidazolium salt precursor and surface-anchored NO₂-NHCs were summarized in Supplementary Table 2.

The asymmetric N-O vibration and aromatic C=C vibration were detected in both the ATR-IR and AFM-IR spectra with relatively small shifts of up to 5 cm⁻¹ in the peak position. The absence of the symmetric N–O vibration in the AFM-IR spectrum can be connected with the fact that AFM-IR measurements are more sensitive to vibrations that are perpendicular to the surface. Thus, the lack of a symmetric N–O vibration can indicate that the –NO₂ groups in EC-deposited NO₂-NHCs were not oriented in a standing position, as identified in DFT calculations (Supplementary Fig. 3) and in other addressable NHC monolayers[17,48].

AFM topography image along with the corresponding AFM-IR mapping at 1533 cm⁻¹ are shown in Fig. 4c, d, respectively. The AFM-IR map reveals homogeneous distribution of the vibrational signal at 1533 cm⁻¹ across the Au surface. No signal was detected on the bare Si surface. AFM-IR mapping at 1603 cm⁻¹ showed a uniform distribution of the vibrational signals on the same area (Supplementary Fig. 14). These results suggest that there is a uniform chemical functionality of surface-anchored NHCs in the

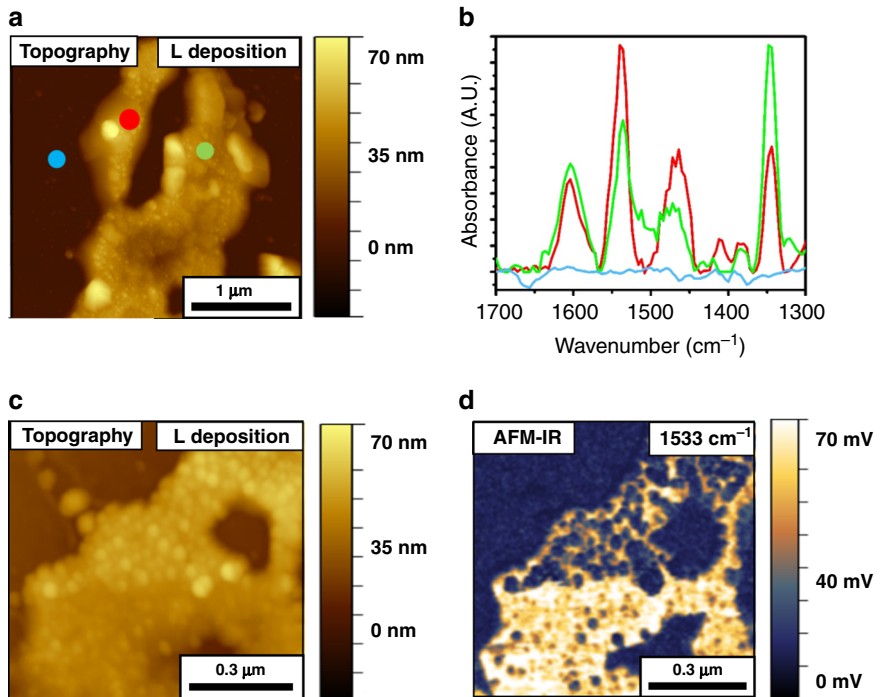

**Fig. 5 AFM-IR measurements of NO$_2$-NHCs that were deposited on Au film by base-induced deprotonation.** AFM topography (**a**) and AFM-IR point spectra measurements (**b**) following base-induced deprotonation deposition of NO$_2$-NHCs on a patchy Au-coated Si substrate. Colored circles in **a** mark the local IR measurement positions and the measured IR spectra are shown in **b** with identical color-coding. Higher magnification AFM topography image and the corresponding AFM-IR image at 1533 cm$^{-1}$ are shown in **c** and **d**, respectively.

mapped surface. It should be noted that the AFM-IR measurements provide averaged nanoscale information about the chemical properties of surface-anchored NHCs over an area of 1 μm$^2$. Analysis of the distribution and chemical functionality of NHCs on metal surfaces at this scale cannot be easily achieved by conducting STM measurements.

The AFM topography image (Fig. 4c) showed randomly distributed structures in the size range of 10–70 nm, which were scattered on both the Au film and Si substrate and were higher by 10–15 nm from their surrounding environment (randomly distributed structures are highlighted in Supplementary Fig. 15). These structures did not show the indicative IR absorption at 1533 cm$^{-1}$ (Fig. 4d). AFM phase image revealed differences between the phase of the randomly distributed structures and their surrounding environment (Supplementary Fig. 15). These structures can be attributed to bromide residues, which were detected by XPS measurements (Supplementary Fig. 2), and locally blocked the NHCs' adsorption on the Au film.

SAM of NO$_2$-NHCs was also prepared on a patchy Au film by base-induced deprotonation and characterized by AFM-IR measurements (Fig. 5). The colored dots in the AFM topography image (Fig. 5a) represent the sites in which localized IR measurements were performed. The corresponding IR spectra were plotted in Fig. 5b with the same color-coding.

AFM-IR spectra showed significant vibrational features at 1346 and 1533 cm$^{-1}$ that correspond to the symmetric and asymmetric N–O vibrations, respectively (Fig. 5b)[17]. A signal at 1466 cm$^{-1}$ was detected and assigned to a C-NH vibration. Vibrational signature was also probed at 1603 cm$^{-1}$ and correlated to aromatic C=C vibrations. No vibrational signature was identified on the bare Si surface, indicating that NO$_2$-NHCs were solely anchored on the Au surface. The selective adsorption of NO$_2$-NHCs on Au surfaces, following base-induced deprotonation, was also previously identified by synchrotron-based IR nanospectroscopy measurements[28].

ATR-IR spectrum of the nitro-functionalized imidazolium salt precursor (Supplementary Fig. 13 and Supplementary Table 2) showed similar peaks to those detected in the AFM-IR spectra. However, the peak at 1466 cm$^{-1}$, which was detected in the AFM-IR spectra and correlated to C-NH vibration, was not probed in the ATR-IR spectra. This result validates that this peak was obtained due to the reduction of –NO$_2$ groups.

The IR peaks at 1346 and 1466 cm$^{-1}$, correlated to symmetric N–O and C-NH vibrations, respectively, which were detected in the AFM-IR spectra of NO$_2$-NHCs that were prepared by base-induced deposition (Fig. 5b), were not detected in the AFM-IR spectra of EC-deposited NO$_2$-NHCs (Fig. 4b). The presence of a vibrational signal at 1466 cm$^{-1}$, which presumably results from the reduction of nitro groups to amines, demonstrates the reductive nature of base-induced deprotonation deposition, as identified by XPS and LSV measurements (Fig. 3c, d). The detection of both the symmetric and asymmetric N-O vibrations in base-induced deposited NHCs can be correlated to the random orientation of the –NO$_2$ groups. Comparison of the AFM-IR amplitudes revealed that EC-deposited NO$_2$-NHCs has 2-fold higher signals than NO$_2$-NHCs that were prepared by base-induced deprotonation (Supplementary Fig. 16). This variation reflects the higher surface density of EC-deposited NHCs.

AFM topography image at higher magnification showed that the Au surface became decorated with nanoparticles in the size range of 10–50 nm following base-induced deposition of NO$_2$-NHCs (Fig. 5c). AFM-IR mapping at 1533 cm$^{-1}$ (Fig. 5d) revealed that while the flat areas on the Au film were characterized with a strong vibrational signal, no vibrational signature was detected on areas that were decorated by nanoparticles. This observation is consistent with the hypothesis that the nanoparticles blocked the NHCs' adsorption sites. These nanoparticles may be constructed of potassium and bromide residues, which their presence on the surface was probed by XPS measurements (Supplementary Fig. 2). AFM phase imaging

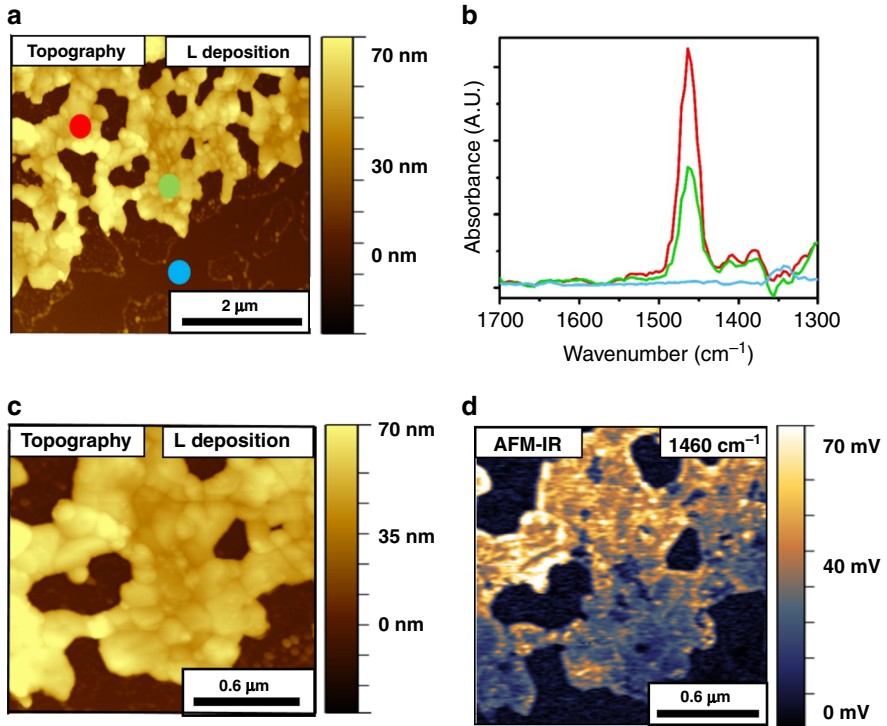

**Fig. 6 AFM-IR measurements of EC-deposited NO$_2$-NHCs on Au film following exposure to one LSV cycle.** AFM topography (**a**) and AFM-IR point spectra (**b**) of EC-deposited NO$_2$-NHCs after one LSV cycle. The vibrational signals were acquired at different locations as indicated by dots with corresponding colors in the AFM topography image. Higher magnification AFM topography image and the corresponding AFM-IR image at 1460 cm$^{-1}$ are shown in **c** and **d**, respectively.

identified as well phase dissimilarities between the nanoparticles and their surrounding Au surface (Supplementary Fig. 17).

AFM topography (Fig. 6a) and AFM-IR measurements (Fig. 6b) of EC-deposited NO$_2$-NHCs were conducted after one LSV cycle (0.15 to −1 V at 0.1 V/s) in order to identify the influence of electroreduction on the vibrational properties of the SAM. AFM-IR measurements showed IR spectra with a single peak at 1463 cm$^{-1}$, corresponding to N–H vibration (Fig. 6b). The lack of N–O signatures in the IR spectra demonstrates the high efficiency of the electroreduction process. No vibrational signatures were detected on the bare Si surface, indicating that electroreduction did not lead to diffusion of NHCs into the Si substrate. The absence of aromatic C=C signal at 1603 cm$^{-1}$ can either reflect that the molecules have changed their orientation into a more flat-lying position or can be the result of the deteriorated surface-density of NHCs due to electrodesorption[17]. AFM topography measurement at higher magnification (Fig. 6c) and AFM-IR mapping of the same area at 1460 cm$^{-1}$ (Fig. 6d) revealed that areas that were adjacent to the Si substrate showed weaker vibrational signals. This result indicates that partial NHCs' desorption from the Au surface, which was facilitated by electroreduction, has mostly occurred on sites that were located in proximity to the Si substrate.

**EC deposition of NO$_2$-NHCs on various metal films**. One of the advantages in the EC deposition approach is that it can be widely utilized for deposition of NHCs on various conductive substrates. To demonstrate this feasibility, NO$_2$-NHCs were EC-deposited, in addition to Au, on Pt, Pd, and Ag films. N1$s$ XPS measurements identified that SAMs of NO$_2$-NHCs were formed by EC-deposition on the various metal films (Fig. 7). Interestingly, an inverse correlation was detected between the atomic ratio of Br and that of N on the various metal surfaces (Supplementary Table 3). This result demonstrates the competitive adsorption of

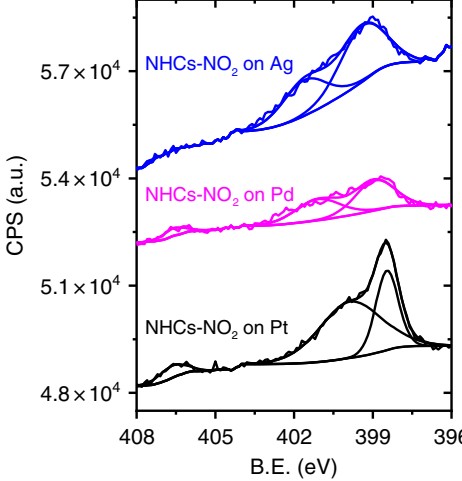

**Fig. 7 N1$s$ XPS measurements of EC-deposited NO$_2$-NHCs on various metal films.** N1s XPS signals of EC-deposited NO$_2$-NHCs on Pt, Pd, and Ag films (black-, magenta- and blue-colored spectra, respectively).

bromide and carbene on the metal surface. The variation in the NO$_2$/NH$_x$ peaks area ratio among the different metals was correlated to differences in their affinity for dissociative chemisorption of H$_2$, which is formed during EC deposition. Thus, a more inert surface toward H$_2$ dissociation, such as Au, led to higher NO$_2$/NH$_x$ ratio.

It should be noted that SAM formation of NHCs on Ag films was not previously reported while using imidazolium salts as precursors. In previous reports the deposition of NHCs on Ag required highly controlled environment (Ultra High Vacuum conditions and cryogenic temperature) and using NHC–CO$_2$

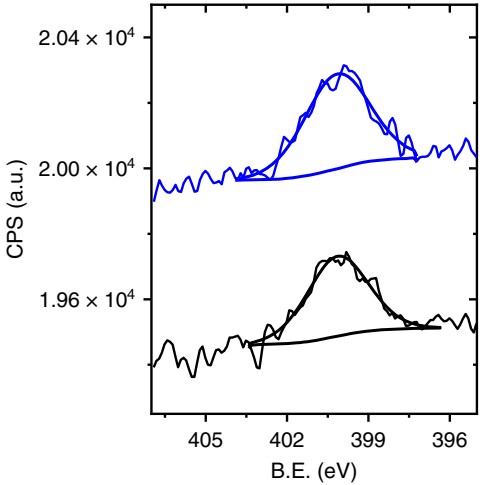

**Fig. 8 N1s XPS signals of EC-deposited dimethyl-benzimidazole on Au film.** EC deposition was conducted with 5 and 25 mM dimethyl-benzimidazolium iodide (black- and blue-colored spectra, respectively).

adduct as precursor[13]. The challenges in surface-anchoring of NHCs on Ag surfaces can be related to the presence of oxidized Ag or to the strong interaction of Ag with halides. It is postulated that the reductive conditions of the EC deposition and the presence of a relatively small concentration of halides during the deposition enabled the formation of a SAM of NHCs on the Ag film by this deposition method.

**EC deposition of a dimethyl-benzimidazole monolayer on Au film.** In order to show the wide applicability of the EC deposition approach, we have expanded our NHCs scope and demonstrate that dimethyl-benzimidazole can be EC-deposited on Au surfaces. N1s XPS spectra were acquired following EC deposition of 5 mM dimethyl-benzimidazolium iodide with 50 mM $H_2O$ at −1 V. The successful EC deposition of dimethyl-benzimidazole on Au was identified by the presence of a single peak in the N1s XPS spectrum (Fig. 8, black-colored spectrum). No significant changes were detected in the peak area once the concentration of dimethyl-benzimidazolium iodide in the EC deposition was increased by 5-fold to 25 mM (Fig. 8, blue-colored spectrum). This result demonstrates that an optimize coverage was already achieved at lower concentration and validates the self-limited EC deposition process that led to monolayer formation.

To conclude, in this work we demonstrate that NHC-based SAMs can be prepared by using electrochemically assisted deprotonation of 2,4 dinitrophenyl-imidazolium bromide and dimethyl-benzimidazolium iodide. In the EC deposition process, hydroxide ions are electrochemically formed on the metal electrode by water reduction. The localized base formation enabled deprotonation of the imidazolium salt precursors and NHCs' anchoring on various metal surfaces under ambient conditions without using an exogenous base during the deposition process. EC-deposited SAMs of $NO_2$-NHCs were characterized with improved chemical stability and higher surface density than SAMs of $NO_2$-NHCs that were prepared by base-induced deprotonation. Moreover, high spatial resolution AFM-IR measurements revealed that the Au surface on which $NO_2$-NHCs were EC-deposited had lower concentration of residues and was characterized with a more homogeneous distribution of NHCs in comparison to a surface on which NHCs were deposited by base-induced deprotonation. These advantages were attributed to the mild conditions under which the EC-induced deprotonation reaction was facilitated and to the fact that a small and constant

concentration of active carbenes was continuously formed in proximity to the metal surface. The wide applicability of the EC deposition approach was demonstrated and SAMs of $NO_2$-NHCs were EC-deposited on Au, Pt, Pd and Ag surfaces. The wide applicability, higher surface density and improved chemical stability of EC-deposited NHC-based SAMs, along with the fact that EC deposition does not require dry conditions or the use of external base, makes it a desirable and easily applicable method for preparation of NHC-based SAMs on metal substrates.

## Methods

**Electrochemical deposition.** Au films (100 nm) were evaporated on a highly doped n-type Si wafer. The Au-coated Si wafers (2 cm × 1 cm) were thoroughly rinsed and dried under nitrogen prior to deposition of NHCs. EC depositions were conducted with a potentiostat (CHI-630, CH Instruments). The EC deposition setup consists of a conventional three-electrode cell, with the metal-coated Si wafer as the working electrode, Ag/AgBr as a quasi-reference electrode and a platinum wire was used as a counter electrode, 5 mM solution of 1,3-*bis*(2,4-dinitrophenyl)-imidazolium bromide salt (prepared according to a previously published protocol)[28] in acetonitrile along with 0.1 M of a supporting electrolyte (Tetrabutylammonium tetrafluoroborate) and 50 mM triple-distilled water at room temperature. A voltage of −1 V was applied for five minutes. After this step, the Au-coated Si wafer was rinsed by three cycles of acetonitrile, triple-distilled water and ethanol, following 5 min flow of $N_2$. A similar procedure was performed for EC deposition of $NO_2$-NHCs on Ag, Pd, and Pt films and for EC deposition of 1,3-dimethyl-benzimidazolium iodide on Au films.

**Base-induced deposition.** 1,3-*bis*(2,4-dinitrophenyl)-imidazolium bromide was prepared and activated in a glove box according to a previously published protocol[28]. The freshly prepared carbene solution was transferred into a vial in which the Au-coated Si wafers were deposited. After 18 h, the wafer was removed from the glove box and rinsed three times with THF (5 ml) and distilled water (5 ml), intermittently. The sample was flushed with $N_2$ for 5 min.

**XPS measurements.** X-ray photoelectron spectroscopy (XPS) measurements were performed using Kratos AXIS Supra spectrometer (Kratos Analytical Ltd., Manchester, U.K.) with Al Kα monochromatic X-ray source (1486.6 eV). The XPS spectra were acquired with a takeoff angle of 90° (normal to analyzer); vacuum condition in the chamber was $2 \times 10^{-9}$ Torr. High-resolution XPS spectra were acquired with a pass energy of 20 eV and step size of 0.1 eV. The binding energies were calibrated according to the Au$4f_{7/2}$ XPS peak position (B.E. = 84.0 eV). Data were collected and analyzed by using ESCApe processing program (Kratos Analytical Ltd.) and Casa XPS (Casa Software Ltd.).

**LSV measurements.** LSV measurements were conducted with a potentiostat (CHI-630, CH Instruments) using a three-electrode glass cell. Ag/AgCl (KCl 1 M) was used as a reference electrode and platinum wire was used as a counter electrode. The samples were immersed in 0.1 M HCl (aqueous) during LSV measurements and the voltage was scanned from 0.15 to −0.5 V at 0.1 V/s.

**AFM-IR measurements.** Au films (100 nm) were evaporated on a highly doped n-type Si wafer and annealed under nitrogen to 300 °C for 10 h for the formation of patchy Au films. These conductive Au films were prepared without exposure to exogenous source of carbon or the use of an adhesive layer. Tapping-mode AFM-IR measurements were performed using a nanoIR3 system (Bruker) equipped with Bruker Hyperspectral QCL laser source (800-1800 cm$^{-1}$). AFM-IR measurements were performed using gold-coated Si probes with a nominal diameter of ~25 nm, resonance frequency values of 75 ± 15 kHz and spring constant values of 1–7 N/m. Averaged spectral acquisition time was 5 s/spectra with resolution of 4 cm$^{-1}$. All spectra were averaged and smoothed using Savitzky-Golay filter.

## Data availability

The authors declare that all the important data to support the findings in this paper are available within the main text or in the supplementary information. Extra data are available from the corresponding author upon reasonable request.

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

## Acknowledgements

This research was supported by the European Research Council (ERC) under the European Union's Horizon 2020 research and innovation program (Grant Agreement No. 802769, ERC Starting Grant "MapCat"). L.D. thanks the Israeli Ministry of Science. S.D. acknowledges the Israeli Ministry of Energy and the Azrieli Foundation for the award of an Azrieli Fellowship. F.D.T. thanks the Director, Office of Science, Office of Basic Energy Sciences and the Division of Chemical Sciences, Geosciences, and Biosciences of the US Department of Energy at LBNL (DE-AC02-05CH11231) for partial support of this work. We acknowledge the assistance of Dr. Netta Bruchiel-Spanier in conducting the electrochemical experiments and Prof. Ori Gidron for insightful discussions.

## Author contributions

E.A., L.D., and S.D. performed the electrochemical experiments and analyzed the spectroscopic data. S.K. and F.D.T synthesized the carbene precursors. A.R and Q.H. performed the AFM-IR measurements. H.E and T.S. conducted theoretical calculations. V.G. performed the XPS measurements. D.M. supervised the electrochemical measurements. S.D. and E.G. conceived and designed the experiments. E.A., S.D., and E.G. wrote the manuscript with help from the other authors. E.G. supervised the project.

## Competing interests

The authors declare no competing interests.
