## [Peer Review File · Nature Communications]

REVIEWER COMMENTS

Reviewer #1 (Remarks to the Author):

A new method is proposed to anchor N-heterolytic carbenes on metal surfaces. This method relies on an electrochemically induced process. It presents advantages compared to the classical chemical methods as it requires less harsh conditions. The paper is easy to follow and the deposited layers are characterized using XPS and AFM-IR techniques. It is shown that the newly presented method leads to higher surface coverage. I think this new technique is very interesting and deserved to be published. However, I think that the mechanism underlying the process (water reduction producing local concentration of hydroxide allowing deprotonation of the imidazolium to form the carbene) is not supported by evidence and is highly speculative. The authors should provide more mechanistic evidence or modify the text and be less affirmative. In the same vein, the claim (in the title) that monolayers are formed seems to be overselling the results: the "monolayer type" deposition is not proved. I would also suggest the authors give information on the passivation (or not) effect of the deposited layers (for example by testing the CV response of a reversible homogeneous couple on a bare and on a modified gold electrode). Finally, the authors only tested the modified electrodes in water; what about the response in an organic solvent where the NO₂ reduction signal could be reversible.

Based on the above comments, I recommend publication with minor revision.

Reviewer #2 (Remarks to the Author):

This paper demonstrates a new preparation approach of attaching N-heterocyclic carbene (NHC) species onto metal surfaces by electrochemical deposition method (EC-deposition method), which achieved a SAM of higher surface density and higher stability than previous preparation method (base-induced deprotonation method), and an application to several metal surfaces. The method is very interesting, and the characterizations of the prepared EC-deposited NHC SAM by XPS, electrochemical analyses, and AFM-IR were fairly convincing by comparing the data with the NHC SAM prepared by base-induced deposition. I have several important questions/comments that need to be addressed before I can recommend acceptance of this paper in Nature Communications.

(1) Overall: The electrochemical deposition method to prepare NHC SAM seems to be a nice method, and the experimental proof of this method is performed rather easily by tracing the fate of NO₂ species in this particular NHC. The reviewer is curious about the scope and limitation of NHCs. Is this method useful for other NHCs? If so, the authors should provide additional experimental data of other NHCs; such as NHCs that are used to prepare SAM by previous methods (ex. base-induced deprotonation). This will strongly support that this method is not specialized to this particular NHC.

(2) Page 5, line 112: The applied potential is -1 V for electrochemical deposition. The authors must show a reference electrode potential, otherwise it is difficult to understand what the reference electrode is. The authors must review the experimental section (Page 6, lines 137-147) and provide necessary information (the model of potentiostat, the types of reference and counter electrode,

etc). During this EC deposition process, water is reduced to form hydroxide ions at the electrode surface, which function as a base for the deprotonation of the imidazolium cation precursors for NO₂-NHCs. If this hydroxide ion generation process is an already reported procedure, the authors must make more appropriate citation (The citation position of references 35-37 seems to be vague.). If this observation is new and specific for the present electrochemical system, then the authors must present the data for hydroxide ion generation.

(3) Page 6, line 1: The authors wrote the used NHC simply as “nitro-functionalized NHC (NO₂-NHC)”, however, the actual chemical structure of the used NO₂-NHC was not present in the manuscript and its name was not clear. The authors must show the chemical structure and the name of the used NO₂-NHC (together with the imidazolium salt precursor: what is the counter anion?) explicitly at the beginning of the manuscript (including experimental section) and supporting information, although the authors have continuously used the same type of NHCs. The information in only Scheme 1 is not enough.

(4) Page 9, line 192: The authors should cite appropriate references for the attribution of NO₂ and C-NH_x species in N 1s XPS.

(5) Page 9, line 199: The authors use HCl aq for linear sweep voltammetry measurement to reduce NO₂ functional group by the electrochemical reduction using proton and electron applied from working electrode, and estimate the number of NO₂-NHC molecules attached on the metal electrode. The objective of this LSV experiment seems not clear in this paragraph; therefore, the authors should review this paragraph again by showing the objective of this experiment and the mechanism of electrochemical reduction together with citing appropriate references. In addition, the authors must review the corresponding experimental section (Page 7, lines 161-166); in what experiment Ag/AgBr quasi-reference electrode was used?

(6) Page 9, line 201, Page 10, line 225: black-colored spectrum → black-colored voltammogram
Linear sweep voltammograms are not spectra. The authors must check the terms throughout the manuscript and SI.

(7) Page 10, line 241: The reviewer cannot find the information in Table S1. The authors must review this sentence.

(8) The AFM-IR experiment of the NO₂-NHCs monolayers provides the spatial distribution of monolayers by monitoring FT-IR spectra; and comparing the data of monolayers prepared by two methods provide the obvious differences. The authors use the FT-IR data of imidazolium salt precursor measured by ATR-IR to attribute the IR spectra in AFM measurement. Why is the FT-IR spectra of NO₂-NHC (after deprotonation) not used for comparison, since this would provide the more direct comparison? The FT-IR spectra of NO₂-NHC (after deprotonation) should be provided, if possible, although it depends on the stability. The reasons why the FT-IR data of imidazolium salt precursor was used for comparison and why ATR method is used must be present in the manuscript explicitly (page 13, lines 312-). The experimental procedure of ATR-IR measurement must also be provided (in the supporting information, for instance).

(9) Page 13, lines 326-330: The magnifications of AFM-IR images at 1533 cm^{-1} (Figure 2d) and 1603 cm^{-1} (Figure S6) are different, which would not make the correct comparison. The images of same magnification must be provided. In detail, the AFM-IR image of 1533 cm^{-1} with the same magnification to Figure 2a and Figure S6; and the AFM-IR image of 1603 cm^{-1} with the same magnification to Figure 2d must be provided.

(10) Page 13, line 377: It is difficult to figure out randomly distributed structures in Figure 2c, and should be modified. Same information seems to be present in Figure S7; therefore, addition of Figure S7 together with Figure 2c in the sentence might be effective.

(11) For the peak attribution of AFM-IR data of NO_2 -NHC monolayers, a summary table of the observed peak positions, simulated peak positions from DFT, and their attributions would be helpful to understand the data.

(12) Page 16, line 370: Was the peak attribution derived from the simulation, or citation? This must be explicit.

Reviewer #3 (Remarks to the Author):

The manuscript entitled "Electrochemical Deposition of Addressable N-Heterocyclic Carbene Monolayers on Metal Surfaces" by Einav Amit et al. reported an interesting work on the field of surface modification. It opens a new way for the elaboration of SAMs by means of the imidazolium electrografting. Authors combine various techniques (electrochemistry, XPS, FTIR, AFM) to analyze the results. Also, DFT calculation was performed to investigate the process. They compared the chemical and electrochemical N-Heterocyclic Carbene SAMs formation, and they showed clearly that the electrochemical deposition is easier. The chosen methodology and the obtained results meet the standard in the field of electrodeposition.

However, I have few comments:

1. Authors summarized in Scheme 1b the electrochemically-induced deprotonation of imidazolium salts precursors for the N-Heterocyclic carbenes monolayer formation. In practice, the solution in which the electrodeposition occurred contained a mixture of acetonitrile and water. However, it is well known that the acido-basic properties in a given solution depend upon the composition of the solution. What is the influence of the ratio acetonitrile/water on the deprotonation and the deposition process ?
2. In Scheme 2, the electrochemical deposition utilizes the localized formation of hydroxide ions in proximity to the electrode surface, induced by water reduction. Also, in order to have an idea of the strength of the acido-basic reaction that takes place in the vicinity of the electrode surface, it will be interesting to give the pKa values of hydrogen between two nitrogen within the imidazolium.

3. Authors reported that this approach should be of interest for many applications: sensors, catalysis,...Can authors give information about the stability issue after continuous cycling in few typical solvents for electrochemical applications (such as aqueous solution and acetonitre) ?

Based on the above comments, I recommend minor revisions before the publication of this manuscript in Nature Communication.

We thank the referees for their comments. We have revised the manuscript to address the referees' comments. The modified manuscript presents a more comprehensive version of our work and provides fundamental insights about the electrodeposition mechanism and expands the EC-deposited NHCs scope. Specifically, the following aspects were modified in the revised version of the manuscript:

1. Analysis of electrodeposition mechanism: additional XPS, UV-Vis absorption spectroscopy and spectro-electrochemistry measurements were performed (Fig. S5-S10) in order to uncover the electrodeposition mechanism. Additionally, passivation measurements were performed to exclude the formation of multilayer by EC-deposition (Fig. S4) and stability measurements of the EC-deposited monolayer were performed (Fig. S11).
2. Wider scope of NHCs: We have demonstrated that EC-deposition can be also applied for surface-anchoring of dimethyl-benzimidazolium iodide as an example for EC-deposition of non-addressable benzimidazolium (Fig. S18).

In the following document we addressed the referees' comments one by one. The referees comments (copied from the original text) are colored in black, while our reply to the referees' comments is colored in blue. Changes or additions that were made in the original manuscript were highlighted in yellow both in this text and in the highlighted version of the manuscript.

Reviewer 1:

A new method is proposed to anchor N-heterolytic carbenes on metal surfaces. This method relies on an electrochemically induced process. It presents advantages compared to the classical chemical methods as it requires less harsh conditions. The paper is easy to follow and the deposited layers are characterized using XPS and AFM-IR techniques. It is shown that the newly presented method leads to higher surface coverage. I think this new technique is very interesting and deserved to be published.

Comment: However I think that the mechanism underlying the process (water reduction producing local concentration of hydroxide allowing deprotonation of the imidazolium to form the carbene) is not supported by evidences and is highly speculative. The authors should provide more mechanistic evidences or modify the text and be less affirmative.

Reply: The following additional experiments were performed in order to provide mechanistic evidence for the role of water reduction and hydroxide ion formation for imidazolium deprotonation and formation of carbenes that can be self-assembled on metal surface.

Spectroelectrochemistry measurements were conducted to demonstrate that both a potential of -1 V and the presence of water are essential for

imidazolium deprotonation and carbene formation. UV-Vis absorption spectra of 2,4 dinitrophenyl-imidazolium bromide salt in acetonitrile showed an absorption peak at 430 nm (black-colored spectrum). Following addition of KO^tBu to the solution, an additional peak at 550 nm was detected and correlated to deprotonation and carbene formation (red-colored spectrum). Applying a negative potential (-1 V) in the absence of water did not lead to carbene formation as identified by the lack of absorption feature in the 550 nm range (magenta-colored spectrum). However, a broad peak was detected in the UV-Vis spectrum (480-650 nm) once 50 mM H₂O was added to the solution and -1 V was applied (blue-colored spectrum). The presence of this broad feature is correlated to EC-induced deprotonation and carbene formation. The similarities in the pattern of the UV-Vis spectra of imidazolium salt that was deprotonated by inorganic base (red-colored spectrum) and by applying -1 V in the presence of water (blue-colored spectrum) validates our hypothesis that water reduction enabled imidazolium deprotonation and carbene formation.

The following sentence was added to the manuscript (page 13) and the following figure (Figure S5) was added to the supporting information:

“Spectroelectrochemistry measurements demonstrated that both negative potential (-1 V) and H₂O addition are essential for imidazolium deprotonation and carbene formation (Figure S5)”

Figure S5. UV-Vis absorption spectra of 1,3-bis(2,4-dinitrophenyl)-imidazolium bromide in acetonitrile before (black-colored spectrum) and following addition of KO^tBu (red-colored spectrum). UV-Vis absorption spectra of 1,3-bis(2,4-dinitrophenyl)-imidazolium bromide in acetonitrile while applying a voltage of -1 V on Au electrode in the absence of water (magenta colored spectrum) and with 50 mM H₂O (blue colored spectrum).

Complementary XPS experiments demonstrated the crucial role of hydroxide ions in surface-anchoring of carbenes. N1s XPS measurements were performed following EC deposition at -1 V without water (blue colored

spectrum) and EC deposition at -0.5 V with 50 mM of water (red colored spectrum). In both experiments no N1s signal was detected, indicating that NHCs were not adsorbed on the surface. N1s XPS signal was detected on the Au surface only when EC deposition was performed at -1 V and 50 mM of water was added to the solution (black-colored spectrum).

The following sentences were added to the manuscript to describe the XPS results (page 13) and the following Figure was added to the supporting information (Figure S6):

“Surface anchoring of NO₂-NHCs was not achieved once the EC deposition was performed in the absence of water or at lower potential (-0.5 V), demonstrating that both a negative potential of -1 V and water addition are essential for EC-deposition of NHCs (Figure S6)”.

Figure S6. N1s XPS spectra of Au electrode following its exposure to 5 mM of 1,3-bis(2,4-dinitrophenyl)-imidazolium bromide in acetonitrile with 50 mM water and while applying -0.5 V (blue-colored spectrum); applying -1 V without water addition (red-colored spectrum); and while applying -1 V with 50 mM of water (black-colored spectrum).

After we demonstrated that hydroxide ions formation is a crucial aspect for enabling the EC deposition, we verified the influence of water concentration on the EC-deposition of NO₂-NHCs by conducting N1s XPS measurements. A lower density of NHCs was detected once the water concentration was decreased by an order of magnitude, from 50 to 5 mM (green-colored spectrum). Higher water concentration did not increase the surface density of NHCs, but led to undesired oxidation of the surface-anchored NHCs (blue-colored spectrum).

The following sentences were added to the main text (page 13) and the following Figure (Figure S7) was added to the supporting information:

“The influence of water concentration on the EC-deposition yield was studied (Figure S7). It was identified that the surface density of NHCs was fourfold lower once water concentration was decreased from 50 to 5 mM. The surface

density of NHCs was not changed once water concentration was increased to 150 mM, demonstrating the self-limited process of monolayer formation by EC-deposition. However, higher water concentration induced undesired oxidation reactions within the surface-anchored NHCs.”

Figure S7. N1s XPS spectra of EC-deposited NO₂-NHCS on Au with various concentration of H₂O in the solution.

The feasibility for EC-deposition is based on the fact that hydroxide ions, which are formed by water electroreduction, will function as a base for deprotonation of the imidazolium salt. Thus, the pKa of the imidazolium salt should be lower than the pH on the electrode. In order to validate this point, we have measured the pKa of 2,4 dinitrophenyl-imidazolium and calculated the pH in the vicinity of the electrode under EC-deposition.

The following paragraph was added to the manuscript (page 14) to describe the pKa measurements and the analysis of the pH in the vicinity of the electrode under EC-deposition:

“The feasibility for EC-deposition is based on the fact that hydroxide ions, which are formed by water electroreduction, will function as a base for deprotonation of the imidazolium salt. The pKa of 2,4 dinitrophenyl-imidazolium was measured (see Supporting Information for additional details and Fig. S8-S10) and was equal to $pK_a = 10.49 \pm 0.02$, while the pH in the vicinity of electrode was estimated to be $pH = 12.54$ (see Supporting Information for additional details). Thus, the pKa of the imidazolium salt is lower than the pH on the electrode during EC-deposition, enabling deprotonation of the imidazolium salt by hydroxide ions.”

The following paragraphs were added to the supporting information:

2. pKa measurements of 2,4 dinitrophenyl-imidazolium and pH analysis in the vicinity of the electrode

The pKa of the 2,4 dinitrophenyl-imidazolium was calculated by titration with KOH in DMSO while monitoring the changes in the carbene concentration by UV-Vis spectroscopy measurements. Fig. S8 shows the absorption spectra of imidazolium salt in DMSO before and after addition of KOH and KO^tBu, demonstrating the similarities in the carbene absorption pattern following deprotonation by the two bases. KOH was used as a base for the titration experiments due to its simple 1:1 stoichiometric ratio in acid-base reaction with the imidazolium. The influence of KOH addition on the UV-Vis absorption spectra of the imidazolium salt solution in DMSO is shown in Fig. S9. Analysis of the peak amplitude at 435 nm as function of the KOH concentration is shown in Fig. S10.

pKa analysis was performed by determining, based on UV-Vis absorption spectra, the amount of KOH that was needed in order to deprotonate half of the imidazolium salt, using the following equation:

$$K = \frac{[\textit{imidazolium}]^+ [\textit{OH}]^-}{[\textit{carbene}]}$$

At half equivalent point of the titration ($A_{1/2} = A_{\text{max}}/2$) the concentrations of carbene and salt are equal and therefore the pKa can be directly analyzed.

Using this method the pKa was calculated with three different initial concentrations of imidazolium salt (10, 20 and 30 μM), yielding:

$$\text{pKa} = 10.49 \pm 0.02$$

The pH near the electrode was calculated based upon the following:

$$\text{Area of the electrode (A)} = 0.35 \text{ cm}^2$$

The distance that the hydroxide ions travel from the electrode depends on their diffusion coefficient (D_{OH}) and the deposition duration (assuming linear diffusion from a rectangular electrode).

$$D_{\text{OH}} = 5.3\text{E-}5 \text{ [cm}^2\text{/sec]} \text{ (diffusion in water)}$$

The electrodeposition duration (t) was 5 min (300 sec):

$$\text{Distance} = \text{sqrt}(D_{\text{OH}} \cdot t) = \text{sqrt}(5.3\text{E-}5 \text{ cm}^2\text{/sec} \cdot 300 \text{ sec}) = 0.126 \text{ cm}$$

The volume of the diffusion layer (V_{OH}) equals the area of the electrode (A) multiplied by the linear diffusion distance of the ions:

$$V_{\text{OH}} = A \cdot \text{Distance} = 0.35 \text{ cm}^2 \cdot 0.126 \text{ cm} = 0.044 \text{ mL}$$

Q (total charge in coulomb) = $i \cdot t = 0.5 \text{ mA} \cdot 300 \text{ sec} = 0.15 \text{ C}$

Dividing in Faraday's constant will yield the number of moles:

$N = Q / F = 0.15 \text{ C} / 96,485 \text{ C/mol} = 1.55 \text{ E-6 mol}$

Now, the concentration of OH^- can be calculated: $C_{\text{OH}^-} = N/V = 1.55\text{E-6 mol} / 0.044 \text{ mL} = 3.53 \text{ E-2 [M]}$

$C_{\text{H}_3\text{O}^+} = 10^{-14} / 3.53 \text{ E-2 [M]} = 2.83 \text{ E-13 M}$

$\text{pH} = -\log(C_{\text{H}_3\text{O}^+}) = 12.54$

Thus, the pH on the electrode is higher than that of the pKa and this result further assures our assumption that the EC-induced deprotonation is facilitated by hydroxide ions.

Figure S8. UV-Vis absorption spectra of 2,4 dinitrophenyl-imidazolium bromide (10 μM) in DMSO before (black-colored spectrum) and following addition of 10 μM KOBu (red-colored spectrum) and KOH (blue colored spectrum).

Figure S9. UV-Vis absorption spectra of 10, 20 and 30 μM 1,3-bis(2,4-dinitrophenyl)-imidazolium bromide in DMSO following addition of 0.01 M KOH (green-, red- and blue-colored spectra, respectively).

Figure S10. Amplitude of the absorption peak at 435 nm as function of KOH concentration with 1,3-bis(2,4-dinitrophenyl)-imidazolium bromide ($10\mu\text{M}$) in DMSO.

To conclude, the additional experiments validated our hypothesis for the EC-deposition reaction mechanism as depicted in Scheme 2:

1. Water and -1 V are necessary for carbene formation and its following surface-anchoring.
2. Monolayer density depends on the amount of added water.
3. The pKa of the 2,4 dinitrophenyl-imidazolium ion is lower than the pH on the electrode, enabling imidazolium deprotonation under -1 V.

Comment: In the same vein, the claim (in the title) that monolayers are formed seems to be overselling the results: the “monolayer type” deposition is not proved.

Reply: See our reply above and additional CV experiments below that verified that monolayer (and not multilayer) is formed by EC deposition.

Comment: I would also suggest the authors give information on the passivation (or not) effect of the deposited layers (for example by testing the CV response of a reversible homogeneous couple on a bare and on a modified gold electrode).

Reply: Following the reviewer’s comments we have measured the CV of $\text{Fe}(\text{CN})_6^{3-}$ on the bare Au electrode and the $\text{NO}_2\text{-NHC}$ coated electrode (black and red-colored voltammograms, respectively) to identify if any passivation was induced due to the presence of the deposited molecules. As demonstrated in Fig. S4, no significant changes were detected in the CV spectra. This result excludes the formation of multilayer by EC-deposition.

The following sentence was added to the manuscript (page 13) to describe the results of these experiments: “Reduction and oxidation cycles of $\text{Fe}(\text{CN})_6^{3-}/\text{Fe}(\text{CN})_6^{4-}$ on the bare and $\text{NO}_2\text{-NHC}$ coated Au electrode showed that no passivation of the electrode was induced following EC-deposition (Supp. Info. Fig. S4), thus excluding multilayer formation by EC-deposition”.

Figure S4. CVs of 2 mM $[\text{Fe}(\text{CN})_6]^{3-}$ in 0.1 M KCl recorded with Au electrode before and following EC-deposition of $\text{NO}_2\text{-NHC}$ s (black- and red-colored voltammogram, respectively).

Comment: Finally, the authors only tested the modified electrodes in water; what about the response in an organic solvent where the NO₂ reduction signal could be reversible.

Reply: There are two main difficulties in testing the oxidation of amines-functionalized NHCs: 1. It requires high (>1 V) positive voltages (Langmuir 1994, 10, 1306-1313) that can also lead to desorption of the monolayer from the surface. 2. It can lead to the formation of various side reactions such as coupling reaction between neighboring molecules or deformation of the surface anchored carbene.

Due to these challenging aspects we have focused our study in the more feasible and less destructive nitro-reduction reaction.

Based on the above comments I recommend publication with minor revision.

We thank the reviewer for the assessment and the insightful comments.

Reviewer 2:

This paper demonstrates a new preparation approach of attaching N-heterocyclic carbene (NHC) species onto metal surfaces by electrochemical deposition method (EC-deposition method), which achieved a SAM of higher surface density and higher stability than previous preparation method (base-induced deprotonation method), and an application to several metal surfaces. The method is very interesting, and the characterizations of the prepared EC-deposited NHC SAM by XPS, electrochemical analyses, and AFM-IR were fairly convincing by comparing the data with the NHC SAM prepared by base-induced deposition. I have several important questions/comments that need to be addressed before I can recommend acceptance of this paper in Nature Communications.

Comment: Overall: The electrochemical deposition method to prepare NHC SAM seems to be a nice method, and the experimental proof of this method is performed rather easily by tracing the fate of NO₂ species in this particular NHC. The reviewer is curious about the scope and limitation of NHCs. Is this method useful for other NHCs? If so, the authors should provide additional experimental data of other NHCs; such as NHCs that are used to prepare SAM by previous methods (ex. base-induced deprotonation). This will strongly support that this method is not specialized to this particular NHC.

Reply: Following the referee's comment and in order to demonstrate the wide applicability of the EC-deposition approach we have expanded our NHCs scope and demonstrated that dimethyl-benzimidazolium can be EC-deposited on Au surfaces.

The following sentences summarize our experimental observations and were added to the manuscript (page 23) along with Fig. S18 that was included in the supporting information:

"In order to show the wide applicability of the EC-deposition approach we have expanded our NHCs scope and demonstrate that dimethyl-benzimidazolium can be EC-deposited on Au surfaces. N1s XPS spectra were acquired following EC-deposition of 5 and 25 mM dimethyl-benzimidazolium iodide with 50 mM H₂O at -1 V. the successful EC-deposition of dimethyl-benzimidazolium monolayer on Au was identified by the presence of a single peak in the N1s XPS spectrum (Supp. Info. Fig. S18). No significant changes were detected in the peak area once the concentration of dimethyl-benzimidazolium iodide was increased by fivefold to 25 mM. This result demonstrates that an optimize coverage was already achieved at lower concentration and validates the self-limited EC-deposition process that led to monolayer formation."

Figure S18. N1s XPS spectra of Au electrodes following exposure to 5 and 25 mM dimethyl-benzimidazolium iodide (black- and blue-colored spectra, respectively) in acetonitrile under applied negative potential of -1 V and 50 mM H₂O.

Comment: Page 5, line 112: The applied potential is -1 V for electrochemical deposition. The authors must show a reference electrode potential, otherwise it is difficult to understand what the reference electrode is. The authors must review the experimental section (Page 6, lines 137-147) and provide necessary information (the model of potentiostat, the types of reference and counter electrode, etc).

Reply: The following details were added to the experimental section (page 6): "Au films (100 nm) were evaporated on a highly doped n-type Si wafer. The Au-coated Si wafers (2 cm x 1cm) were thoroughly rinsed and dried under nitrogen prior to deposition of NHCs. EC depositions were conducted with a potentiostat (CHI-630, CH Instruments). The EC-deposition setup consists of a conventional three-electrode cell, with the metal-coated Si wafer as the working electrode, Ag/AgBr as quasi-reference electrode and a platinum wire was used as a counter electrode."

Comment: During this EC deposition process, water is reduced to form hydroxide ions at the electrode surface, which function as a base for the deprotonation of the imidazolium cation precursors for NO₂-NHCs. If this hydroxide ion generation process is an already reported procedure, the authors must make more appropriate citation (The citation position of references 35-37 seems to be vague.). If this observation is new and specific for the present electrochemical system, then the authors must present the data for hydroxide ion generation.

Reply: The formation of a locally high pH by water electroreduction was previously utilized by us and others for facilitating various reactions, such as

sol-gel formation. The following reference (#37) was added to the manuscript to demonstrate the utilization of hydroxide ions for facilitating chemical process: "Shacham, R., Avnir, D. and Mandler, D., 1999. Electrodeposition of Methylated Sol-Gel Films on Conducting Surfaces. *Advanced Materials*, 11(5), pp.384-388".

Comment: Page 6, line 1: The authors wrote the used NHC simply as "nitro-functionalized NHC (NO₂-NHC)", however, the actual chemical structure of the used NO₂-NHC was not present in the manuscript and its name was not clear. The authors must show the chemical structure and the name of the used NO₂-NHC (together with the imidazolium salt precursor: what is the counter anion?) explicitly at the beginning of the manuscript (including experimental section) and supporting information, although the authors have continuously used the same type of NHCs. The information in only Scheme 1 is not enough.

Reply: The chemical structure of the imidazolium salt precursor and the counter Br⁻ are shown in Scheme 1. In order to clarify the name of the imidazolium salt the following sentences were added to the manuscript:

Page 6: "1,3-bis(2,4-dinitrophenyl)-imidazolium bromide salts were used as a model system for addressable carbenes and were EC-deposited on various metal surfaces."

Scheme 1 legend: "by electrochemically-induced deprotonation of imidazolium 1,3-bis(2,4-dinitrophenyl)-imidazolium bromide salts precursors.

Page 6: "The EC-deposition setup consists of a conventional three-electrode cell, with the metal-coated Si wafer as the working electrode, 5 mM solution of 1,3-bis(2,4-dinitrophenyl)-imidazolium bromide the imidazolium salt in acetonitrile along with 0.1 M of a supporting electrolyte (Tetrabutylammonium tetrafluoroborate) and 50 mM triple-distilled water at room temperature."

Conclusions: "In this work we demonstrate a new approach for NHC-based SAM formation by using electrochemically-assisted deprotonation of 2,4-dinitrophenyl-imidazolium bromide and dimethyl-benzimidazolium iodide."

Comment: Page 9, line 192: The authors should cite appropriate references for the attribution of NO₂ and C-NH_x species in N 1s XPS.

Reply: The following reference (#41) was added to the manuscript for attribution of the N1s XPS signals: "Graf, N.; Yegen, E.; Gross, T.; Lippitz, A.; Weigel, W.; Krakert, S.; Terfort, A.; Unger, W. E. S. XPS and NEXAFS studies of aliphatic and aromatic amine species on functionalized surfaces. *Surf. Sci.* 2009, 603, 2849– 2860"

Comment: Page 9, line 199: The authors use HCl aq for linear sweep voltammetry measurement to reduce NO₂ functional group by the electrochemical reduction using proton and electron applied from working electrode, and estimate the number of NO₂-NHC molecules attached on the metal electrode. The objective of this LSV experiment seems not clear in this paragraph; therefore, the authors should review this paragraph again by showing the objective of this experiment and the mechanism of electrochemical reduction together with citing appropriate references.

Reply: Following the referee's comment we have added the following sentence and references to the manuscript (Page 10):

“Electroreduction of the nitro groups in surface-anchored NO₂-NHCs provides a chemical handle for quantitative analysis of the surface density of NHCs, based on the well-documented mechanism of electroreduction of aromatic nitro compounds.^{42,43} “

The following references (#42, #43) were added to the manuscript:

- Cadle, S. H., Tice, P. R. & Chambers, J. Q. Electrochemical Reduction of Aromatic Nitro Compounds in Presence of Proton Donors. *J Phys Chem* **71**, 3517, doi:DOI 10.1021/j100870a025 (1967).
- Futamata, M., Nishihara, C. & Goutev, N. Electrochemical reduction of p-nitrothiophenol-self-assembled monolayer films on Au(111) surface and coadsorption of anions and water molecules. *Surf Sci* **514**, 241, doi:Pii S0039-6028(02)01636-9

Comment: In addition, the authors must review the corresponding experimental section (Page 7, lines 161-166); in what experiment Ag/AgBr quasi-reference electrode was used?

Reply: We have revised the experimental section and the following sentence was re-written (page 7):

“LSV measurements were conducted with a potentiostat (CHI-630, CH Instruments) using a three-electrode glass cell. Ag/AgCl (KCl 1 M) was used as a reference electrode and platinum wire as a counter electrode.

Comment: Page 9, line 201, Page 10, line 225: black-colored spectrum →black-colored voltammogram. Linear sweep voltammograms are not spectra. The authors must check the terms throughout the manuscript and SI.

Reply: We have changed the term “black-colored spectrum” into “black-colored voltammogram” throughout the text.

Comment: Page 10, line 241: The reviewer cannot find the information in Table S1. The authors must review this sentence.

Reply: The sentence in Page 11 was modified from “(Supp-Info, Table S1)” to “(details about the quantitative analysis can be found in the supporting information)”

Comment: The AFM-IR experiment of the NO₂-NHCs monolayers provides the spatial distribution of monolayers by monitoring FT-IR spectra; and comparing the data of monolayers prepared by two methods provide the obvious differences. The authors use the FT-IR data of imidazolium salt precursor measured by ATR-IR to attribute the IR spectra in AFM measurement. Why is the FT-IR spectra of NO₂-NHC (after deprotonation) not used for comparison, since this would provide the more direct comparison? The FT-IR spectra of NO₂-NHC (after deprotonation) should be provided, if possible, although it depends on the stability. The reasons why the FT-IR data of imidazolium salt precursor was used for comparison and why ATR method is used must be present in the manuscript explicitly (page 13, lines 312-). The experimental procedure of ATR-IR measurement must also be provided (in the supporting information, for instance).

Reply: ATR-IR spectrum of the imidazolium salt precursor was included in the supplementary information as a reference point to enable the comparison between the IR spectrum of the precursor and the AFM-IR spectrum of the surface-anchored molecule.

The FTIR spectrum of the surface anchored NO₂-NHC on Au was previously reported by us (Fig. S1 in *Chemistry–A European Journal*, 25, 66, 2019, 15067) and therefore was not included in this study.

Figure S1 in *Chemistry–A European Journal*, 25, 66, 2019, 15067. IRRAS spectrum of the as-deposited NO₂-functionalized NHCs on Au (111).

The following sentence was added to the manuscript to discuss the connection between the ATR-IR and the previously reported IRRAS measurements (page 16): “Infrared reflection absorption spectrum of surface-anchored NO₂-NHCs showed peaks at similar positions to those detected by ATR-IR.¹⁷ⁿ”

The following sentence was added to the supporting information to describe the ATR-IR setup: “Attenuated total reflectance Fourier transform infrared (ATR-FTIR) measurements of 1,3-bis(2,4-dinitrophenyl)-imidazolium bromide were made using a Thermo Scientific Nicolet iS50 instrument and a diamond ATR crystal”.

Comment: Page 13, lines 326-330: The magnifications of AFM-IR images at 1533 cm^{-1} (Figure 2d) and 1603 cm^{-1} (Figure S6) are different, which would not make the correct comparison. The images of same magnification must be provided. In detail, the AFM-IR image of 1533 cm^{-1} with the same magnification to Figure 2a and Figure S6; and the AFM-IR image of 1603 cm^{-1} with the same magnification to Figure 2d must be provided.

Reply: Figure S14 was modified so that the AFM topography, AFM-IR mapping at 1603 cm^{-1} and AFM-IR mapping 1533 cm^{-1} will be presented at the same magnification.

Figure S14: AFM topography and AFM-IR mapping at 1603 and 1533 cm^{-1} of EC-deposited NO_2 -NHCs.

Comment: Page 13, line 377: It is difficult to figure out randomly distributed structures in Figure 2c, and should be modified. Same information seems to be present in Figure S7; therefore, addition of Figure S7 together with Figure 2c in the sentence might be effective.

Reply: The following sentence (page 17) was added to the manuscript to connect Figure 2c and Figure S15. In addition, the randomly distributed structures were highlighted by green circles in Fig. S15:

“The AFM topography image (Fig. 2c) showed randomly distributed structures in the size range of 10-70 nm, which were scattered on both the Au film and Si substrate and were higher by 10-15 nm from their surrounding environment (randomly distributed structures are highlighted in Supp. Info. Fig. S15).”

Figure S15: AFM topography, phase and AFM-IR mapping (at 1533 cm^{-1}) of EC-deposited NHCs. Green circles highlight the appearance of randomly distributed structures in the AFM topography image and green lines connect the appearance of these structures in the three images.

Comment: For the peak attribution of AFM-IR data of NO₂-NHC monolayers, a summary table of the observed peak positions, simulated peak positions from DFT, and their attributions would be helpful to understand the data.

Reply: The following sentence was added to the main text (page 16): “The vibrational peaks of the imidazolium salt precursor and surface anchored NO₂-NHCs were summarized in Table S2.”

The following table was added to the supporting information:

Table S2: IR signals for imidazolium salt precursor and surface-anchored NO₂-NHC

Molecule	Imidazolium salt precursor		NO ₂ -NHC (EC deposition)	NO ₂ -NHC (Base-activated)	NO ₂ -NHC (Base-activated) ^a
	ATR-IR	DFT	AFM-IR	AFM-IR	IRRAS
Symmetric N=O	1340 cm ⁻¹	1368 cm ⁻¹	-	1346 cm ⁻¹	1347 cm ⁻¹
N-H	-	-	-	1466 cm ⁻¹	-
Asymmetric N=O	1536 cm ⁻¹	1562 cm ⁻¹	1536 cm ⁻¹	1533 cm ⁻¹	1544 cm ⁻¹
C=C	1608 cm ⁻¹	1645 cm ⁻¹	1608 cm ⁻¹	1603 cm ⁻¹	1615 cm ⁻¹

^a Values extracted from Chem. Eur. J. 25, 2019, 15067

Comment: Page 16, line 370: Was the peak attribution derived from the simulation, or citation? This must be explicit.

Reply: a citation was added to the following sentence (page 18): “AFM-IR spectra showed significant vibrational features at 1346 and 1533 cm⁻¹ that correspond to the symmetric and asymmetric N-O vibrations, respectively (Fig. 3b).¹⁷”

We thank the reviewer for the assessment and insightful comments.

Reviewer 3:

The manuscript entitled "Electrochemical Deposition of Addressable N-Heterocyclic Carbene Monolayers on Metal Surfaces" by Einav Amit et al. reported an interesting work on the field of surface modification. It opens a new way for the elaboration of SAMs by means of the imidazolium electrografting. Authors combine various techniques (electrochemistry, XPS, FTIR, AFM) to analyze the results. Also, DFT calculation was performed to investigate the process. They compared the chemical and electrochemical N-Heterocyclic Carbene SAMs formation, and they showed clearly that the electrochemical deposition is easier. The chosen methodology and the obtained results meet the standard in the field of electrodeposition.

However, I have few comments:

Comment: Authors summarized in Scheme 1b the electrochemically-induced deprotonation of imidazolium salts precursors for the N-Heterocyclic carbenes monolayer formation. In practice, the solution in which the electrodeposition occurred contained a mixture of acetonitrile and water. However, it is well known that the acido-basic properties in a given solution depend upon the composition of the solution. What is the influence of the ratio acetonitrile/water on the deprotonation and the deposition process?

Reply: Following the referee's comment the influence of water concentration on the EC-deposition of NO₂-NHCs was studied. A lower density of NHCs was detected once the water concentration was decreased by an order of magnitude, from 50 to 5 mM (green-colored spectrum). Higher water concentration did not increase the surface density of NHCs, but led to undesired oxidation of the surface-anchored NHCs (blue-colored spectrum).

The following sentences were added to the main text (page 13) and the following Figure (Figure S7) was added to the supporting information:

"The influence of water concentration on the EC-deposited yield was studied (Figure S7). It was identified that the surface density of NHCs was fourfold lower once water concentration was decreased from 50 to 5 mM. The surface density of NHCs was not changed once water concentration was increased to 150 mM, demonstrating the self-limited process of monolayer formation by EC-deposition. However, higher water concentration induced undesired oxidation reactions within the surface-anchored NHCs. The results of these experiments validate our hypothesis that water reduction led to the formation of a basic environment that facilitated imidazolium deprotonation and the following surface-anchoring of carbene."

Figure S7. N1s XPS spectra of EC-deposited NO₂-NHCS on Au with various concentration of H₂O in the solution.

Comment: In Scheme 2, the electrochemical deposition utilizes the localized formation of hydroxide ions in proximity to the electrode surface, induced by water reduction. Also, in order to have an idea of the strength of the acido-basic reaction that takes place in the vicinity of the electrode surface, it will be interesting to give the pKa values of hydrogen between two nitrogen within the imidazolium.

Reply:

The feasibility for EC-deposition is based on the fact that hydroxide ions, which are formed by water electroreduction, will function as a base for deprotonation of the imidazolium salt. Thus, the pKa of the imidazolium salt should be lower than the pH on the electrode. In order to validate this point, we have measured the pKa of 2,4 dinitrophenyl-imidazolium and calculated the pH in the vicinity of the electrode under EC-deposition. The following paragraph was added to the manuscript to describe the pKa measurements and the analysis of the pH in the vicinity of the electrode under EC-deposition:

"The feasibility for EC-deposition is based on the fact that hydroxide ions, which are formed by water electroreduction, will function as a base for deprotonation of the imidazolium salt. The pKa of 2,4 dinitrophenyl-imidazolium was measured (see Supporting Information for additional details and Fig. S8-S10) and was equal to $pK_a = 10.49 \pm 0.02$, while the pH in the vicinity of electrode was estimated to be $pH = 12.54$ (see Supporting Information for additional details). Thus, the pKa of the imidazolium salt is

lower than the pH on the electrode during EC-deposition, enabling deprotonation of the imidazolium salt by hydroxide ions.”

The following paragraphs were added to the supporting information:

The following paragraphs were added to the supporting information:

2. pKa measurements of 2,4 dinitrophenyl-imidazolium and pH analysis in the vicinity of the electrode

The pKa of the 2,4 dinitrophenyl-imidazolium was calculated by titration with KOH in DMSO while monitoring the changes in the carbene concentration by UV-Vis spectroscopy measurements. Fig. S8 shows the absorption spectra of imidazolium salt in DMSO before and after addition of KOH and KO^tBu, demonstrating the similarities in the carbene absorption pattern following deprotonation by the two bases. KOH was used as a base for the titration experiments due to its simple 1:1 stoichiometric ratio in acid-base reaction with the imidazolium. The influence of KOH addition on the UV-Vis absorption spectra of the imidazolium salt solution in DMSO is shown in Fig. S9. Analysis of the peak amplitude at 435 nm as function of the KOH concentration is shown in Fig. S10.

pKa analysis was performed by determining, based on UV-Vis absorption spectra, the amount of KOH that was needed in order to deprotonate half of the imidazolium salt, using the following equation:

$$K = \frac{[\text{imidazolium}]^+ [\text{OH}]^-}{[\text{carbene}]}$$

At half equivalent point of the titration ($A_{1/2} = A_{\text{max}}/2$) the concentrations of carbene and salt are equal and therefore the pKa can be directly analyzed. Using this method the pKa was calculated with three different initial concentrations of imidazolium salt (10, 20 and 30 μM), yielding:

$$\text{pKa} = 10.49 \pm 0.02$$

The pH near the electrode was calculated based upon the following:

$$\text{Area of the electrode (A)} = 0.35 \text{ cm}^2$$

The distance that the hydroxide ions travel from the electrode depends on their diffusion coefficient (D_{OH}) and the deposition duration (assuming linear diffusion from a rectangular electrode).

$$D_{\text{OH}} = 5.3\text{E-}5 \text{ [cm}^2\text{/sec]} \text{ (diffusion in water)}$$

The electrodeposition duration (t) was 5 min (300 sec):

$$\text{Distance} = \text{sqrt}(D_{\text{OH}} \cdot t) = \text{sqrt}(5.3\text{E-}5 \text{ cm}^2\text{/sec} \cdot 300 \text{ sec}) = 0.126 \text{ cm}$$

The volume of the diffusion layer (V_{OH}) equals the area of the electrode (A) multiplied by the linear diffusion distance of the ions:

$$V_{\text{OH}} = A \cdot \text{Distance} = 0.35 \text{ cm}^2 \cdot 0.126 \text{ cm} = 0.044 \text{ mL}$$

$$Q \text{ (total charge in coulomb)} = I \cdot t = 0.5 \text{ mA} \cdot 300 \text{ sec} = 0.15 \text{ C}$$

Dividing in Faraday's constant will yield the number of moles:

$$N = Q / F = 0.15 \text{ C} / 96,485 \text{ C/mol} = 1.55 \text{ E-6 mol}$$

$$\text{Now, the concentration of } \text{OH}^- \text{ can be calculated: } C_{\text{OH}^-} = N/V = 1.55\text{E-6 mol} / 0.044 \text{ mL} = 3.53 \text{ E-2 [M]}$$

$$C_{\text{H}_3\text{O}^+} = 10^{-14} / 3.53 \text{ E-2 [M]} = 2.83 \text{ E-13 M}$$

$$\text{pH} = -\log(C_{\text{H}_3\text{O}^+}) = 12.54$$

Thus, the pH on the electrode is higher than that of the pKa and this result further assures our assumption that the EC-induced deprotonation is facilitated by hydroxide ions.

Figure S8. UV-Vis absorption spectra of 2,4 dinitrophenyl-imidazolium bromide (10 μM) in DMSO before (black-colored spectrum) and following addition of 10 μM KOBu (red-colored spectrum) and KOH (blue colored spectrum).

Figure S9. UV-Vis absorption spectra of 10, 20 and 30 μM 1,3-bis(2,4-dinitrophenyl)imidazolium bromide in DMSO following addition of 0.01 M KOH (green-, red- and blue-colored spectra, respectively).

Figure S10. Amplitude of the absorption peak at 435 nm as function of KOH concentration with 1,3-bis(2,4-dinitrophenyl)imidazolium bromide (10 μM) in DMSO.

Comment: Authors reported that this approach should be of interest for many applications: sensors, catalysis. Can authors give information about the stability issue after continuous cycling in few typical solvents for electrochemical applications (such as aqueous solution and acetonitrile)?

Reply: The stability of the EC-deposited monolayer following its exposure to 25 and 50 cycles of cyclic voltammogram (-0.5 V to 1 V) was studied in order to address the comment raised by the referee. The following sentences and figure were added (page 14) to address the stability issue of EC-deposited NO₂-NHCs on Au electrode:

“The stability of the EC-deposited NO₂-NHCs was studied following exposure to 25 and 50 cycles of cyclic voltammetry (-0.5 V to 1 V vs Hg/Hg₂SO₄). N1s XPS measurements did not reveal noticeable changes in the surface density of NO₂-NHCs after 25 cycles (Supp. Info. Fig. S11). However, the surface density was fivefold lower after 50 cycles, indicating that electro-induced desorption has occurred.”

Figure S11. N1s XPS spectra of Au electrodes following NO₂-NHCs deposition (black-colored spectrum), and after 25 and 50 cyclic voltammograms (CV) cycles (green and orange colored spectra). CV conditions: 0.1 M KNO₃ aqueous solution, reference electrode - Hg/Hg₂SO₄, counter electrode - Pt, scan range -0.5 V to 1 V, scan rate 0.1 V/sec.

Based on the above comments, I recommend minor revisions before the publication of this manuscript in Nature Communication.

We thank the reviewer for the assessment and insightful comments.

REVIEWERS' COMMENTS

Reviewer #1 (Remarks to the Author):

My previous concerns have been addressed.

Reviewer #2 (Remarks to the Author):

The authors are appreciated for carefully addressing the review comments. The manuscript was greatly improved. I have one comment/suggestion before publication.

(1) The authors demonstrated that the electrochemical deposition method is also applicable for dimethyl benzimidazolium. I would suggest the inclusion of this NHC in the Scheme 1, too. I would also make a suggestion to move Figure S18 to the main text, since the scope of NHC is also mentioned as well as the type of substrates. A comment on the limitation of NHCs for the electrochemical deposition method would be appreciated, if you have tried several NHCs.

Reviewer #3 (Remarks to the Author):

The revised manuscript takes into account all the remarks from reviewers. The responses of Authors are clear and convincing.

I agree for the publication of the revised version.

We thank the referees for their comments. We have revised the manuscript to address the comments of referee #2.

Reviewer #1:

My previous concerns have been addressed.

We thank the reviewer for the assessment.

Reviewer #2:

The authors are appreciated for carefully addressing the review comments. The manuscript was greatly improved. I have one comment/suggestion before publication.

Comment: The authors demonstrated that the electrochemical deposition method is also applicable for dimethyl benzimidazolium. I would suggest the inclusion of this NHC in the Scheme 1, too.

Reply: Scheme 1 was modified to include the electrochemical deposition of benzimidazolium.

a. Previous works

b. This work

Addressable Imidazolium: $\text{R} = 2,4\text{-dinitrophenyl}$, $\text{R}' = \text{H}$, $\text{X} = \text{Br}^-$
Benzimidazolium: $\text{R} = \text{Me}$, $\text{R}' = -\text{C}_4\text{H}_4^-$, $\text{X} = \text{I}^-$
TBATFB = Tetrabutylammonium tetrafluoroborate

Comment: I would also make a suggestion to move Figure S18 to the main text, since the scope of NHC is also mentioned as well as the type of substrates. A comment on the limitation of NHCs for the electrochemical deposition method would be appreciated, if you have tried several NHCs.

Reply: Fig S18 was transferred from the supporting information to the main text (Fig. 8 in the revised version). We are currently studying the scope and limitations of the electrochemical deposition. These results, which are beyond the scope of this work, will be published in a different manuscript.

Figure 8: N1s XPS signals of EC-deposited dimethyl-benzimidazole on Au film. EC-deposition was conducted with 5 and 25 mM dimethyl-benzimidazolium iodide (black- and blue-colored spectra, respectively).

We thank the reviewer for the assessment and the insightful comments.

Reviewer #3:

The revised manuscript takes into account all the remarks from reviewers. The responses of Authors are clear and convincing.

I agree for the publication of the revised version.

We thank the reviewer for the assessment.